# Multilabel prediction of virus target proteins via multimodal graph representation learning

**Kuang Ma, Kaiyu Liu, Yuhui Xin, Rong Liu** ⓘ *

Hubei Key Laboratory of Agricultural Bioinformatics, College of Informatics, Huazhong Agricultural University, Wuhan, People's Republic of China

* liurong116@mail.hzau.edu.cn

## Abstract

Identification of virus target proteins (VTPs) is crucial for understanding viral pathogenesis. Existing computational studies have addressed this issue by predicting host-virus protein interactions, typically framed as a single-label problem. However, targets can be identified using only intrinsic information of host proteins. Moreover, a host protein may participate in the infection processes of multiple viruses, a scenario that can be treated as a multilabel prediction problem. Herein, we present MultiVTP, a multilabel framework for VTP prediction that employs graph learning with multimodal information. This algorithm samples subgraphs centered on query proteins to capture topological properties, while multimodal features are extracted to represent proteins from complementary perspectives. A graph transformer integrates and upgrades these attributes, followed by a progressive layered extraction module that captures both shared and virus-specific binding patterns to predict VTPs. Ablation experiments reveal that graph-based attributes and modules are the key contributors to performance, with additional components leading to further improvements in accuracy. Comprehensive evaluations demonstrate that MultiVTP not only surpasses various baseline models but also remains robust under limited training data. Applying our approach to the human proteome enables the systematic identification of novel VTPs for both individual and multiple viruses.

## Author summary

Existing studies frame the identification of virus target proteins (VTPs) as a single-label prediction task by predicting human-virus protein interactions. In this work, we present a multilabel method by leveraging graph learning and multimodal information. We comprehensively extract network topological properties and multimodal features for protein representation, and adopt a graph Transformer coupled with a progressive layered extraction module to upgrade these feature descriptors for multilabel prediction tasks. We find that graph-based

**Data availability statement:** All data sets, source code, and models in this study can be downloaded from the following link: https://github.com/hzau-liulab/MultiVTP.

**Funding:** This work was supported by the National Natural Science Foundation of China (32071249 to RL). The funders had no role in study design, data collection and analysis, decision to publish, or preparation of the manuscript.

**Competing interests:** The authors have declared that no competing interests exist.

attributes and modules play a crucial role in our model, while auxiliary components further boost prediction accuracy. Our algorithm surpasses baseline models across diverse experimental scenarios, especially in cases with limited training samples. We finally apply our algorithm to the human proteome, enabling the systematic identification of novel VTPs associated with both individual and multiple viruses.

## Introduction

Human-virus protein–protein interactions (PPIs) are involved in various biological processes that viruses hijack for their replication [1]. Virus target proteins (VTPs) are host proteins specifically recognized and bound by viral components. In host cells, these proteins not only participate in immune regulation but also orchestrate multiple stages of the viral life cycle [2,3]. For instance, the human protein angiomotin promotes human immunodeficiency virus 1 (HIV-1) budding by bridging the viral Gag protein and the host protein NEDD4L [4]. Identifying VTPs is essential for elucidating the molecular mechanisms of viral pathogenesis and provides promising avenues for antiviral drug development. Leveraging experimental techniques such as yeast two-hybrid assays (Y2H) and affinity purification mass spectrometry (AP-MS), a great number of virus-host protein interactions have been characterized [5,6]. For example, Jäger et al. constructed a high-quality HIV-human protein interactome using AP-MS [7]. Shapira et al. identified host factors involved in influenza virus replication through Y2H [8]. Gordon et al. generated a global protein interaction map of SARS-CoV-2 using AP-MS, while Stukalov et al. profiled the interactomes of SARS-CoV-2 and SARS-CoV to perform a multi-omics analysis [9,10]. Collectively, these studies provide critical experimental datasets for developing computational methods to identify VTPs.

Over the past two decades, extensive efforts have been dedicated to addressing this issue through the prediction of host-virus PPIs. Early approaches primarily relied on machine learning with handcrafted features. For instance, Tastan et al. developed a random forest model that leveraged the functional similarity between human and HIV-1 proteins, gene expression profiles during HIV-1 infection, and topological features of the PPI network [11]. Cui et al. employed a support vector machine (SVM) to predict PPIs involved in hepatitis C virus (HCV) infection using the tripeptide frequency of protein sequences [12]. Dyer et al. represented protein domains as one-hot vectors and used an SVM to recognize interactions between HIV-1 and human proteins [13]. Khorsand et al. developed a random forest model that integrated sequence composition, Gene Ontology (GO) semantic similarity, and network properties to predict PPIs between human and influenza A H1N1 virus [14]. Beyond traditional feature engineering, advanced protein embedding methods have also been applied in this field. The LSTM-PHV model, for example, used Word2Vec for protein sequence embedding and developed a long short-term memory network to capture long-distance dependencies for human-virus PPI prediction [15]. DeepViral utilized

DL2vec to generate embeddings from disease phenotypes and protein functional data, which were then fed into a convolutional neural network (CNN) [16]. Yang et al. integrated embeddings from Doc2vec (sequence), Net2vec (network), and Go2vec (function) into a LightGBM classifier, achieving superior performance in predicting PPIs between human and human papillomavirus (HPV) [17]. More recently, the application of deep learning and pre-trained protein language models has led to notable improvements in prediction performance. Yang et al. devised a transfer learning approach by training a Siamese CNN on source human-virus PPIs using sequence profiles and then fine-tuning the model for other viruses with limited samples [18]. HBFormer introduced a hybrid attention mechanism that integrated sequence embeddings from the ProtTrans language model with biological annotations [19]. The DeepGNHV model incorporated sequence embeddings and AlphaFold2-predicted structures, processing them through a graph neural network (GNN) for human-virus PPI prediction [20]. Collectively, these methods have significantly advanced the development of VTP prediction.

Despite the progress achieved in existing research, several issues warrant further investigation. First, previous methods typically required viral protein information to predict the interaction between a host protein and a virus. In fact, such interactions could be inferred solely based on whether the host protein exhibits characteristics similar to those of known VTPs. To our knowledge, only the HIVPRE algorithm relied exclusively on host protein information to predict interaction propensity with HIV-1 [21]. Second, a human protein could be involved in the infection processes of multiple viruses [22,23]. However, current prediction algorithms were typically designed as single-label binary classification problems for specific viruses. This cannot directly evaluate whether a host protein interacts with multiple viruses, a scenario that could be framed as a multilabel problem. Third, although the human PPI network has been adopted to extract relevant features of host proteins for above tasks, its potential as the basis for a specialized VTP prediction framework remains unexplored [24,25]. Finally, computational identification of potential VTPs for a broad range of viruses within the human proteome may represent a notable gap in the existing studies. Addressing this gap could yield novel insights into the principles governing host-virus interactions.

Inspired by these challenges, we propose MultiVTP, a multilabel model for VTP prediction that integrates graph representation learning with multimodal information. Within this framework, we first sampled multiple subgraphs centered on query proteins to capture their local contexts. For each subgraph, we extracted both global and local network properties to characterize the topological role of every node. Additionally, we incorporated multimodal information, including traditional, sequence, and functional features, to obtain a comprehensive representation of each protein. A graph transformer was then employed to integrate and refine the topological and multimodal features from the subgraphs. Finally, we introduced a specialized module to model both shared and specific binding patterns across multiple viruses, enabling effective multilabel VTP prediction. Extensive evaluations show that MultiVTP outperformed various baseline models and maintained robustness with limited training data. Applied to the human proteome, our approach could enable systematic identification of novel VTP candidates for both individual and multiple viruses.

## Results

### Overview of MultiVTP

Fig 1 illustrates the framework of MultiVTP for predicting the multilabel of VTPs. Our algorithm comprised four stages, including subgraph sampling, feature extraction, feature integration and multilabel prediction. In the first stage, since VTPs generally possess community preference, the neighborhood of a query protein may offer valuable information. We thus sampled a fixed-size subgraph centered on each query protein using the random walk algorithm and repeated this procedure to obtain multi-view subgraphs to capture different neighborhood contexts [26]. In the second stage, we extracted two types of network features from the PPI network for each subgraph, namely the global and local topological properties. The former was calculated by the node2vec algorithm, and the latter was the shortest path distances between nodes [27]. In addition, we extracted multimodal features to characterize the proteins, comprising the traditional, sequence, and functional representations. In the third stage, multimodal features and global topological properties were concatenated and

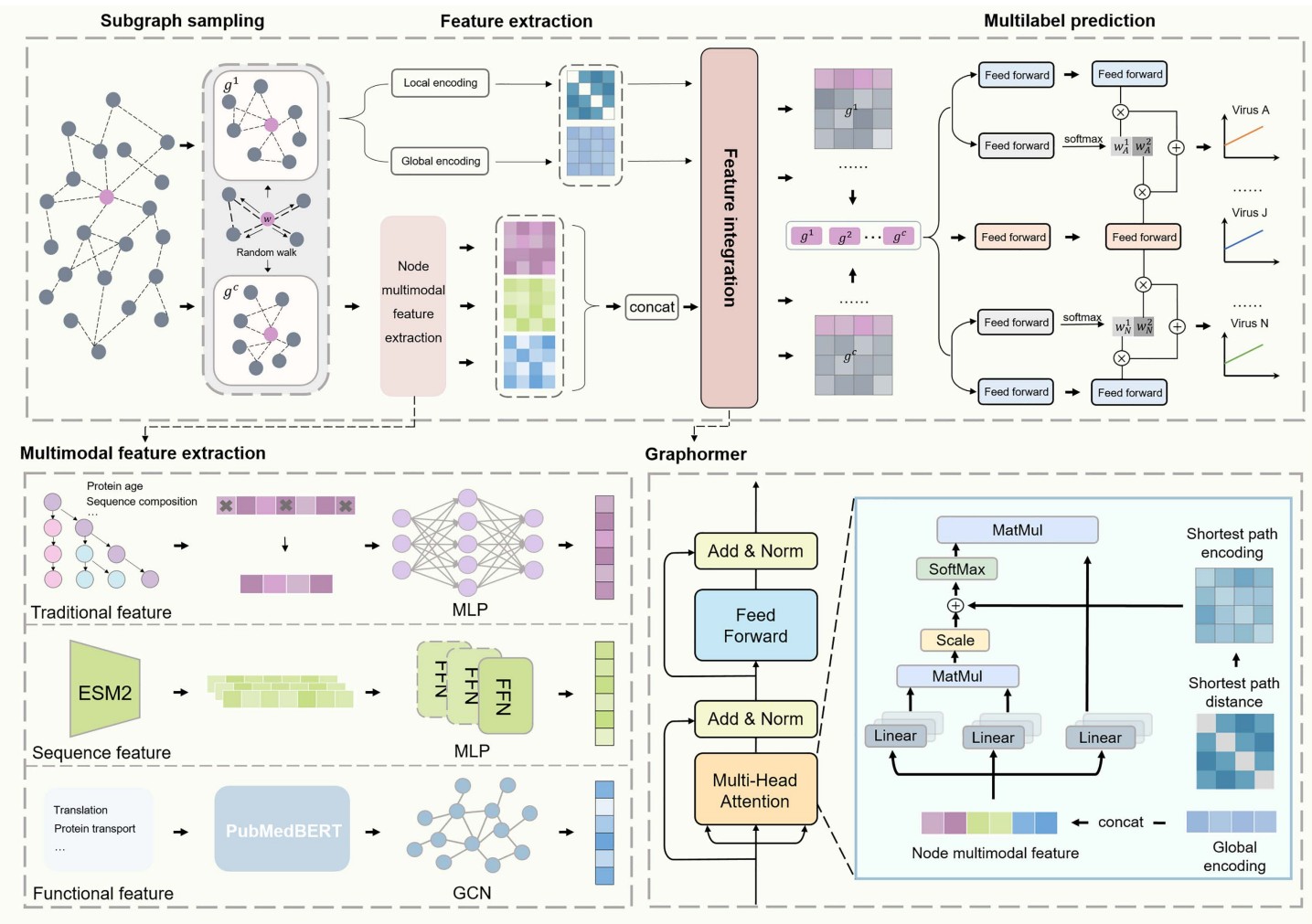

**Fig 1. Framework of MultiVTP.** Our algorithm comprises four stages, including the subgraph sampling, feature extraction, feature integration, and multilabel prediction. First, multi-view subgraphs are sampled from the PPI network for each protein. Second, both network topological (i.e., global and local) properties and multimodal (i.e., traditional, sequence, and functional) features are extracted to represent proteins. Third, the above features are integrated and upgraded using Graphormer within the subgraphs. Finally, a progressive layered extraction model is adopted to capture shared and virus-specific binding signatures, yielding prediction scores for each query protein.

processed through the self-attention mechanism in the Graphormer module, while local topological properties were used to optimize the attention score [28]. This enabled our model to simultaneously focus on the network topology and multi-modal attributes of nodes in the subgraph. In the fourth stage, we used the progressive layered extraction (PLE) model to implement the multilabel prediction. This module generated shared and specific representations using the merged embedding of different subgraphs. Finally, specific classifiers adopted both representations to estimate the labels of query proteins [29].

## Integrative analysis of selected features in this study

In this section, we systemically analyzed the traditional features of samples in the $D_{virus}$ dataset from the sequence, evolutionary, structural and network perspectives (Figs 2A-2D and A in S3 File). In terms of amino acid composition, VTPs

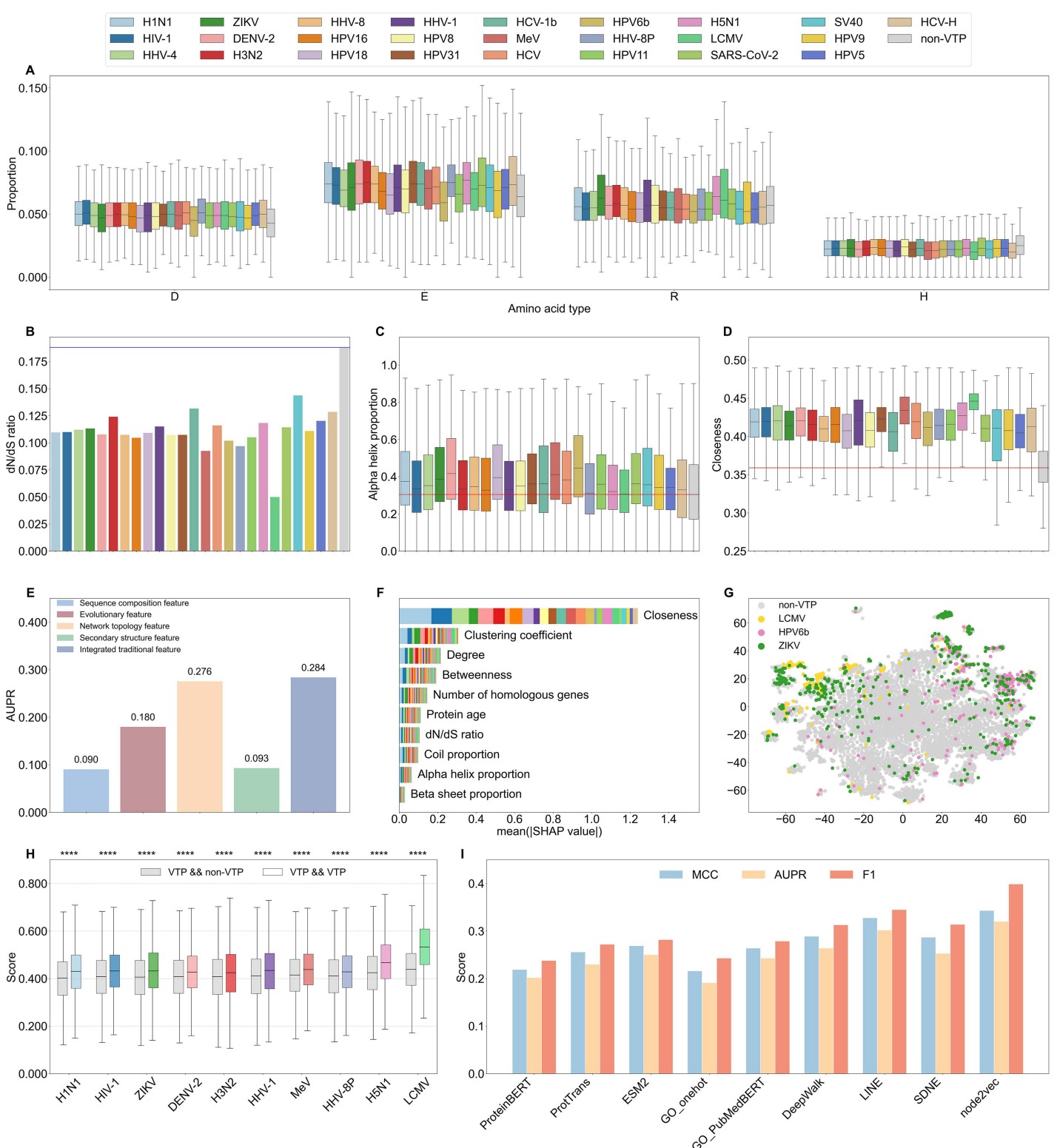

**Fig 2. Feature analysis and comparison on the $D_{virus}$ dataset.** Statistical analysis of traditional features: (A) amino acid composition, **(B)** dN/dS ratio, (C) predicted coil proportion, and (D) closeness. **(E)** AUPRs achieved by different types of traditional features. **(F)** SHAP analysis of traditional features. (G) t-SNE visualization of global topological properties for VTPs and non-VTPs. **(H)** Gene ontology similarity among VTPs and between VTPs and

non-VTPs for each virus. Significant differences are evaluated using Wilcoxon rank sum tests. **** $p < 0.0001$, *** $0.0001 \leq p < 0.001$, ** $0.001 \leq p < 0.01$, * $0.01 \leq p < 0.05$, and ns: $p \geq 0.05$. **(I)** Performance of different protein embeddings.

contained more polar amino acids with negative charges (aspartate and glutamate) and positive charges (lysine) but had a smaller fraction of histidine residues, likely because viral proteins interact with host proteins via electrostatic attraction [30]. In particular, lymphocytic choriomeningitis virus (LCMV) targets exhibited the highest lysine content, whereas HCV-H targets possessed the lowest histidine content (Fig 2A). Compared to non-VTPs, additionally, target proteins showed stronger evolutionary conservation, as indicated by their lower dN/dS ratios, older protein ages, and more homologous genes, implying that more ancient proteins may play a critical role in the infection process. Among the different types of VTPs, LCMV targets were the most evolutionarily conserved (Figs 2B and A in S3 File). Regarding the secondary structure, VTPs had more α-helices and fewer coils than non-VTPs. This may be because viruses favor well-structured and conserved domains over flexible and disordered regions to achieve specific and stable binding to hosts [31–33]. Notably, the targets of Dengue virus 2 (DENV-2) and HPV6b exhibited the highest α-helix content and the lowest coil content, reflecting their potential structural specificity for interacting with the virus (Figs 2C and A-B in S3 File). Based on the human PPI network, VTPs showed higher degree centrality, closeness centrality, clustering coefficient, and betweenness centrality, suggesting that these proteins may be highly connected nodes and occupy more important positions in the network. Compared with other VTPs, LCMV targets showed a more remarkable tendency (Figs 2D and A in S3 File). Moreover, we performed the same analysis for the $D_{family}$ dataset. The similar trends were observed for target proteins with respect to virus families (Figs C-E in S3 File). These results implied that traditional features could distinguish VTPs from non-VTPs and further differentiate VTPs across virus species/families.

We then evaluated the effectiveness of traditional features on the training set using multilayer perceptron (MLP) classifiers. As shown in Fig 2E, the network topological features yielded the highest area under the precision-recall curve (AUPR), followed by evolutionary features, while the sequence and structural attributes had relatively lower measures. Additionally, integration of four types features led to an improved AUPR of 0.284, highlighting their complementarity in the multilabel prediction task. We then used the shapley additive explanations (SHAP) method for feature attribution analysis [34]. The closeness centrality was the most influential feature, while the remaining network topological features also had higher rankings. This suggested the importance of network properties in the identification of VTPs (Figs 2F and F in S3 File). Thus, we used the advanced node2Vec algorithm to capture the global topological properties for each protein. Through t-SNE dimensionality reduction, we observed the separation between VTPs and non-VTPs as well as that among VTPs of different virus species to some extent (Figs 2G and G in S3 File) [35]. VTPs within the same cluster were likely to occupy similar topological positions in the PPI network, suggesting that they could be involved in related biological processes [36]. We then calculated the functional similarity among VTPs and that between VTPs and non-VTPs of each virus using the gosemsim program [37]. The results demonstrated that the VTPs of each virus exhibited higher functional associations, implying that protein functions may provide valuable information for identifying VTPs (Figs 2H and H in S3 File). Based on this, we leveraged the pretrained PubMedBERT model to extract semantic knowledge from GO text descriptions to predict VTPs [38,39]. Finally, we evaluated several widely used protein embeddings from the sequence, functional, and network perspectives. Fig 2I shows that the embeddings of ESM2, PubMedBERT and node2vec performed better than other representations in the corresponding category. Notably, node2vec embedding achieved the optimal performance, owing to its capacity to capture both local community structures and global topological positions. We also combined different deep learning models with these embeddings. Comparative results showed that MLP was optimal for ESM2 embeddings, whereas GNN was more suitable for functional embeddings (Table E in S2 File) [40]. The analysis across virus families is provided in Figs C, F, and I in S3 File and Table E in S2 File. Based on the above results, we propose MultiVTP, a prediction model that adopts Graphormer to integrate these features and the PLE module for extracting

PLOS Computational Biology

valuable information across diverse virus species and families. Excluding viral information may enable the prediction task to concentrate on host susceptibility and pathway-level signals rather than actual host-virus interactions. To verify whether our model identifies genuine interactions, we examined its prediction scores of distinct protein categories. The results indicated that VTPs achieved higher prediction scores than other proteins, verifying that our model effectively captures intrinsic biological signals associated with VTPs (Sections S1.1 and S1.2 in S1 File).

**Interpretability and ablation studies of our model**

To evaluate the contribution of each feature and module in MultiVTP, we conducted ablation studies on training set using 5-fold cross-validation. At the feature level, excluding multimodal features (i.e., only reserving network properties) resulted in reduced AUPR and Matthews correlation coefficient (MCC), which were 0.362 and 0.369, respectively (Fig 3A). Among the multimodal features, traditional features played a more critical role than sequence and functional embeddings. However, these features had different utilities for various virus species. For instance, functional features provided more clues for detecting HPV9 targets, traditional features performed more favorably in identifying HPV31 targets, and sequence features proved more useful for predicting HCV-H targets (Fig J in S3 File). Integrating multimodal information could improve the prediction performance of our model across diverse viruses. However, because we only mitigated redundancy at the sequence level, proteins sharing highly similar functional and topological properties may still lead to overestimated performance. To address this issue, we quantified sample similarity in cross-validation datasets and re-evaluated prediction performance under different similarity thresholds. Our results verified that the utility of MultiVTP was not dependent on redundant information within these features (Section S1.3 in S1 File). At the module level, the removal of Graphormer caused the most obvious performance degradation, where AUPR and MCC values were reduced from 0.389 and 0.394 to 0.304 and 0.326, respectively. Replacing the PLE module with an MLP exerted a negative impact on performance as well. Due to the importance of Graphormer, we performed additional ablation experiments on its components (Fig 3B). Removing global topological properties led to obviously degraded performance, with AUPR and MCC values dropping to 0.327 and 0.340, while removing local topological properties had a marginal impact on performance. Given the importance of network features, we conducted additional ablation studies focusing on the network-related components (Sections S1.4 and S1.5 in S1 File). Moreover, Graphormer used a self-attention mechanism to evaluate the relationship between proteins in the subgraph. When we replaced this model with graph convolutional network (GCN) and graph attention network (GAT), the AUPR decreased to 0.377 and 0.382, respectively, which may be attributed to their difficulty in capturing long-range dependencies. Additionally, experiments with simplified architectures confirmed that each component in the original model contributes to the overall performance (Section S1.6 in S1 File).

Because the use of Graphormer and PLE cooperatively enhanced VTP prediction, we explored the interpretability of these modules using the test set. The self-attention played a crucial role in Graphormer, so we investigated which proteins were assigned higher attentions. For each protein, we calculated its attention score by averaging the attention values assigned to it by other proteins in relevant subgraphs. As shown in Fig 3C, VTPs had higher attention values than non-VTPs. We conducted functional enrichment analysis using DAVID on the top 50% of proteins after sorting them by their attention scores. Fig 3D displays the top 20 enriched biological processes, including host-virus interaction, innate immunity, and antiviral defense [41]. The results indicated that Graphormer could allocate high attentions to virus associated proteins in the self-attention procedure. Furthermore, through the analysis of attention weights between proteins, we found that our model could effectively capture key regulatory mechanisms. Thus, the enhanced attention focused on VTPs and their regulatory partners may reflect the molecular perturbations induced during viral infection (Section S1.7 in S1 File). Following Graphormer, the PLE module generated both shared and specific representations, which were integrated for multilabel prediction. For the shared representation, we assessed its potential in capturing common patterns of VTPs. Thus, we extracted Graphormer's input, Graphormer's output, and shared representation. For each type of embedding, we performed dimensionality reduction using t-SNE. The silhouette coefficient (SC) was used to quantify the degree of

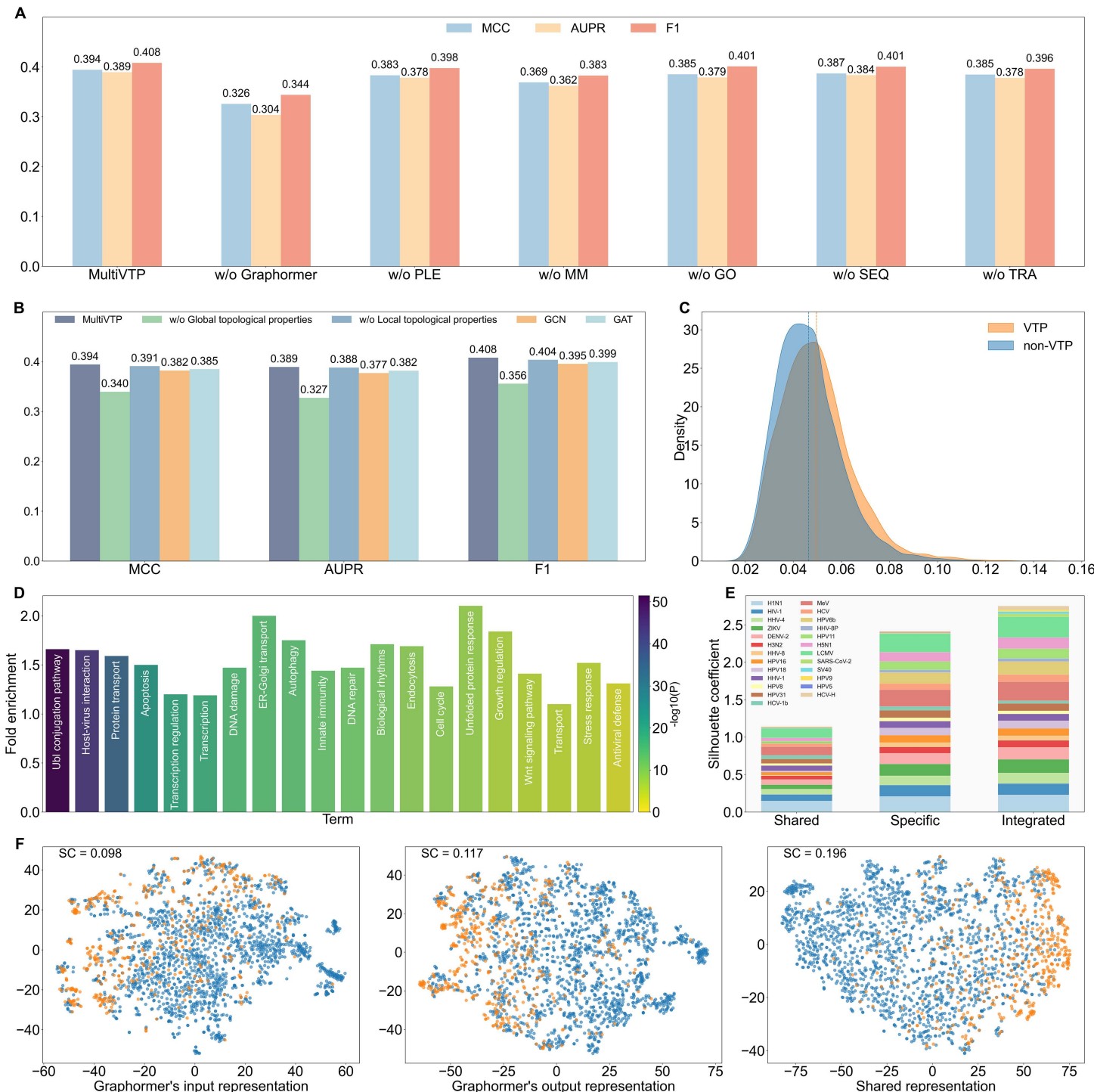

**Fig 3. Interpretability and ablation studies on the $D_{virus}$ dataset. (A)** Ablation studies at both feature and module levels. MM: multimodal features, GO: functional features, SEQ: sequence features, and TRA: traditional features. **(B)** Ablation experiments for the components of Graphormer. **(C)** Density distribution of attention values of VTPs and non-VTPs. **(D)** Top 20 biological processes enriched in the top 50% of proteins sorted by attention scores. **(E)** Silhouette coefficients for evaluating the separation of different samples through t-SNE reduction. (F) t-SNE visualization of representations of VTPs (orange points) and non-VTPs (blue points).

separation among distinct sample types. Compared with the other embeddings, the shared representation more effectively separated VTPs from non-VTPs, with a SC value of 0.198 (Fig 3F). For specific representations, we assessed their capacity to capture the unique patterns of VTPs for each virus. Thus, the SC measure of VTPs and non-VTPs of each virus was computed using the shared embedding, specific embedding and integrated embedding, respectively. Clearly, the specific embedding outperformed the shared representation in separating VTPs from non-VTPs. Moreover, the two embedding types complemented each other by capturing information from different aspects (Fig 3E). The analysis of virus families is presented in Figs J-K in S3 File.

## Advantage of MultiVTP over baseline methods

Most existing methods predicted virus-host interactions based on both viral and host proteins. In contrast, our approach only used the information of host proteins to address this issue. To our knowledge, only the HIVPRE algorithm employed a similar strategy to predict HIV-1 targets using SVMs with different feature combinations [21]. Thus, we compared MultiVTP with three versions of HIVPRE for identifying HIV-1 targets in the test set. Our algorithm outperformed the best model of HIVPRE, with improvements of 0.058 in area under the receiver operating characteristic curve (AUC) and 0.150 in AUPR (Fig 4A-4B). In contrast to our host-only approach, we employed traditional virus-host PPI prediction methods for VTP prediction. However, these methods exhibited a higher tendency to misclassify host proteins as false positives and thus may not be the optimal choice for this task (Section S1.8 in S1 File). To further demonstrate the advantage of MultiVTP, we evaluated several machine learning methods (e.g., MLP, XGB, RF, and SVM) using test set. The MLP generated a numerical vector, each element of which represents the probability of being a virus-specific target. The other machine learning methods were implemented using three multilabel learning strategies, including the binary relevance (BR), classifier chains (CC), and label powerset (LP). Specifically, BR treats each label as an independent binary classification task, LP transforms the problem into a multiclass classification task, and CC models label dependencies by chaining binary classifiers (Fig L in S3 File) [42]. Fig 4C shows that MultiVTP yielded the optimal performance, outperforming XGB-BR and MLP by 0.061 and 0.069 in MCC, and by 0.051 and 0.050 in AUPR, respectively. Compared with XGB, SVM and RF showed more stable performance across the three strategies. Moreover, we evaluated the AUPR in terms of targets of each virus. MultiVTP had better performance for all viruses except HPV16, HCV and HPV5. Notably, although there were only 197 training targets for LCMV, our approach achieved an AUPR of 0.728 and surpassed the second-best method by 0.100 (Fig 4D). A similar trend was observed for the measles virus (MeV), which had 256 training samples. These results implied the potential of our model in few-shot learning tasks. Moreover, we randomly removed a portion of VTPs for each virus from the training set. When 50% of training samples were removed, our model exhibited a slight decrease in AUPR from 0.375 to 0.354. In contrast, the AUPRs of XGB-BR and MLP decreased from 0.324 and 0.325 to 0.283 and 0.263, respectively (Fig 4E). This experiment suggested the robustness of MultiVTP in scenarios with limited training data.

Inspired by the above results, we attempted to check whether our model could be applied to the virus having limited VTPs. To this end, we collected a few-shot learning dataset composed of virus species with 20–100 target proteins. To prevent inflated performance caused by the data leakage, we only retained samples absent from the training set of $D_{virus}$. Finally, the dataset comprised 1,372 non-VTPs and 387 VTPs across 10 virus species (Table D in S2 File). Here, we explored two training strategies: fine-tuning the pre-trained model for new virus and training the model from scratch. The performance was evaluated using stratified 5-fold cross-validation due to limited data. Based on the second strategy, MultiVTP outperformed XGB-BR across all viruses in AUPR, highlighting the superior performance of our model in few-shot learning (Fig 4F). Further, the fine-tuning strategy significantly enhanced prediction performance. For example, our fine-tuned model yielded an AUPR of 0.288 for AAV2, compared to 0.098 for the scratch-trained model, demonstrating the effectiveness of MultiVTP adapted to a novel virus with limited targets. In addition, we constructed more challenging few-shot testing scenarios, which further verified the reliability of our model under data-scarce and emerging virus conditions (Sections S1.9 and S1.10 in S1 File).

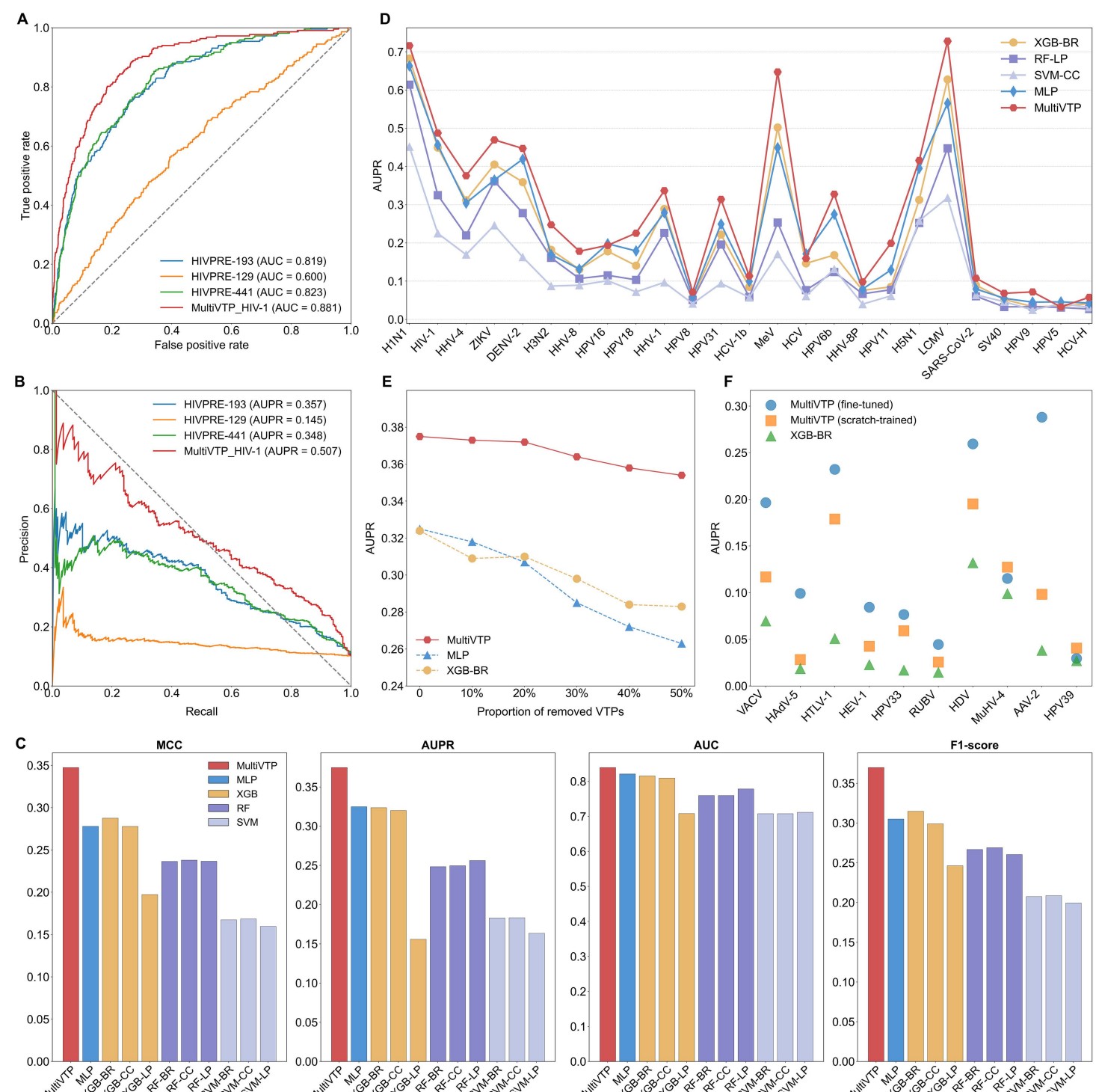

**Fig 4. Performance comparison of MultiVTP and baseline methods on the $D_{virus}$ dataset. (A)** ROC curves of MultiVTP and HIVPRE for HIV-1 target prediction. **(B)** Precision-recall curves of MultiVTP and HIVPRE for HIV-1 target prediction. **(C)** Performance of MultiVTP and machine learning methods. BR: binary relevance, CC: classifier chains, and LP: label powerset. **(D)** Comparison of AUPR between MultiVTP and baseline methods (i.e., MLP and other machine learning models with the optimal learning strategy). Viruses are sorted from left to right in descending order of VTP counts. **(E)** Performance of various approaches after removing different fractions of training VTPs. **(F)** Few-shot evaluation of scratch-trained and fine-tuned MultiVTP.

## Applying MultiVTP to the human proteome

The above experiments indicated that MultiVTP could effectively identify host targets for different viruses. Accordingly, we applied our model to the human proteome, which comprises 20,270 proteins derived from UniProt [43]. The high prediction scores generated by our model may indicate novel targets. For instance, KDELR1 (UniProt ID: P24390), which is not present in virus-host interaction databases, was predicted to interact with DENV-2 with a score of 0.531. This protein shuttles between endoplasmic reticulum and Golgi apparatus and has been reported to interact with the virus [44]. We thus selected the top-ranked proteins as candidates based on the proportion of known targets among the entire human proteome. Fig 5A shows the numbers of novel, recovered, and missing VTPs. Among the viruses, influenza A (H1N1) and HIV-1 had the highest number of candidates (2,555 and 1,681, respectively), of which 1,747 and 923 were recovered VTPs. We then performed GO enrichment analysis on the predicted and known VTPs separately using the DAVID database. As shown in Fig 5B, the candidate proteins shared a substantial number of functional terms with known VTPs, while also possessing distinct functional attributes.

Furthermore, we performed a case study on the predicted VTPs of H1N1. Within the human PPI network, novel VTPs showed a greater tendency to interact directly with known VTPs than non-VTPs (Fig 5C). This implied that the newly predicted targets are likely to play a role in H1N1 virus-host interactions. For instance, the novel VTP YTHDC1 (UniProt ID: Q96MU7) has 142 neighboring proteins, 69 of which are VTPs, and has been experimentally verified to interact with the NS1 protein of H1N1. The high connectivity of novel candidate ICAM1 (UniProt ID: P05362) to known VTPs (145 of 238 neighbors) suggests its role in the early immune response to H1N1 [45]. Existing studies reported that its expression is induced by the VTP IL6 (UniProt ID: P05231), and it then activates another VTP NFKB1(UniProt ID: P19838) to inhibit H1N1 replication [46]. Moreover, KEGG analysis indicated that candidate proteins were enriched in most known VTP-related pathways and associated with 10 novel pathways (Figs 5D and N in S3 File). Notably, six pathways have been experimentally validated as relevant to H1N1 infection (Table G in S2 File). Regarding the adherens junction pathway, for instance, H1N1 infection impairs the epithelial barrier by disrupting adherens junctions, leading to an increase in epithelial permeability and therefore facilitating viral entry [47]. This pathway displayed high interconnectivity among 36 candidate proteins, including 19 recovered VTPs and 8 novel VTPs that were manually validated through literature mining (Fig 5E). The cytoskeletal regulator RAP1B (UniProt ID: P61244), implicated in the life cycles of multiple viruses, may serve as a core factor in this network [48]. Moreover, the analysis of HIV-1 candidates yielded findings similar to those of H1N1. These candidates were enriched in 15 novel KEGG pathways, the majority of which were cancer-associated. This may underlie the molecular mechanisms contributing to the elevated cancer risk observed in HIV-1-infected patients (Fig O in S3 File). In addition, we collected the host dependency factors (HDFs) of HIV-1 and performed a comparative analysis with candidate VTPs. The results indicated that these two types of host proteins were highly associated, and our model was also effective in identifying potential HDFs (Section S1.11 in S1 File).

After analyzing the candidates of individual viruses, we focused on the predicted targets of multiple viruses. Fig 5F shows that our model struggled to recover SVTPs, possibly due to the specific virus-targeting patterns. However, as the number of viruses increased, the number of missing labels decreased progressively. For instance, only 44 labels were missing for proteins targeted by 12 viruses. To investigate the performance discrepancy between single virus target proteins (SVTPs) and multiple virus target proteins (MVTPs), we re-evaluated our model on these two groups separately using 5-fold cross validation. The result indicated that MultiVTP could better capture cross-virus interaction patterns (Fig P in S3 File). The novel MVTPs shared properties with known counterparts, including higher evolutionary conservation and more important topological positions in the PPI network (Fig 5G). Then, we performed GO enrichment analysis for these targets. The top 10 enriched processes, particularly those involved in transcriptional regulation and protein transport, underscored their critical role in the viral life cycle (Fig 5H). For instance, within the protein transport process, the non-VTP SEC61A2 (UniProt ID: Q9H9S3), which was predicted to interact with H1N1, HIV-1, and DENV-2, has been confirmed to be hijacked by these viruses to facilitate the transport of viral proteins [49]. Based on these findings, novel MVTPs hold

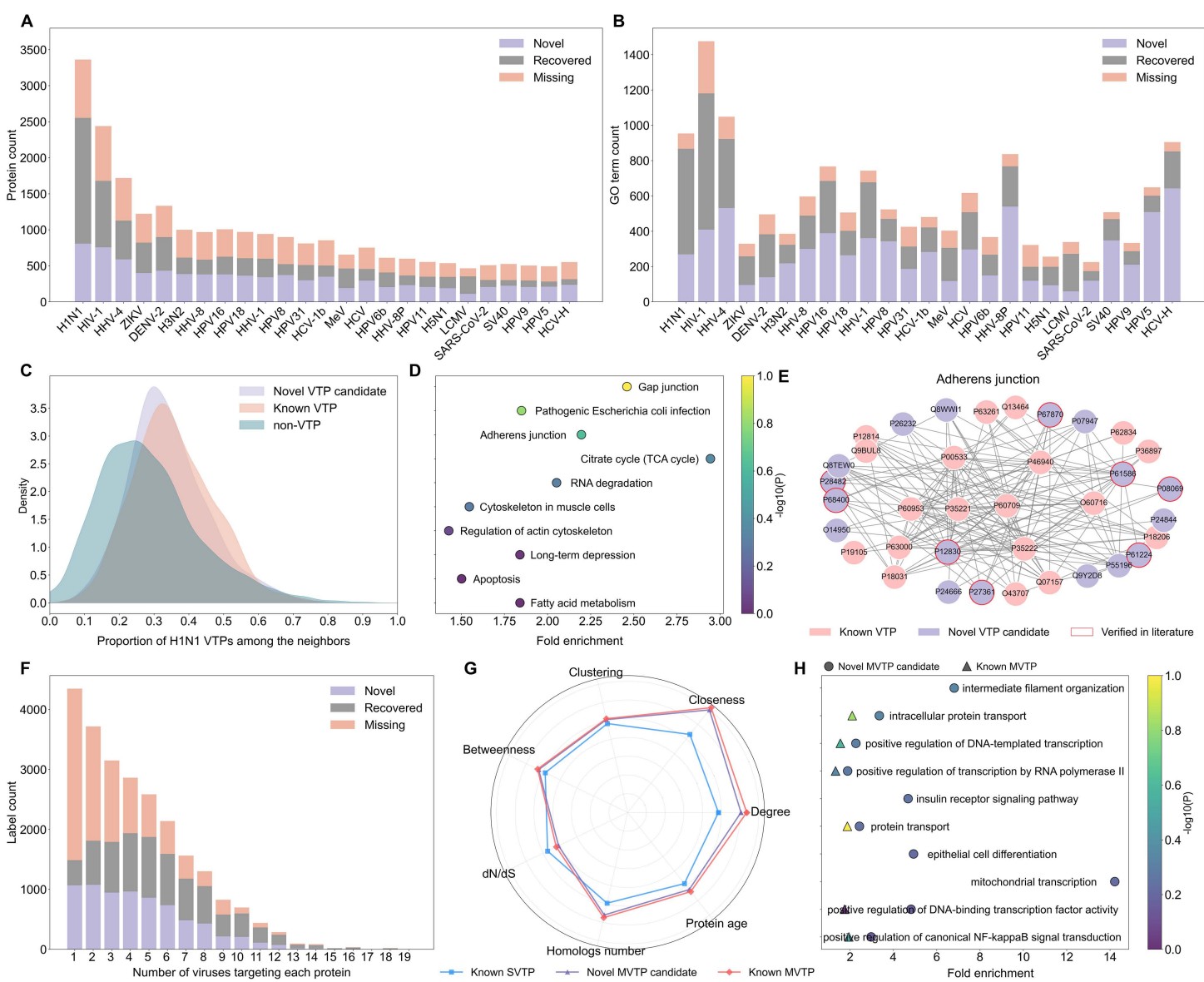

**Fig 5. Analysis of predicted and known VTPs in the human proteome across virus species. (A)** Distribution of predicted and known VTPs for each virus species. The results are classified into three categories: novel (newly predicted VTPs), recovered (overlap between predicted and known VTPs), and missing (known but unrecognized VTPs). **(B)** Distribution of GO terms enriched in predicted and known VTPs for each virus species. The results are classified into three categories: novel (terms enriched exclusively in predicted VTPs), recovered (overlapping terms between predicted and known VTPs), and missing (terms enriched exclusively in known VTPs). **(C)** Density map of proportion of H1N1 VTPs among the interacting neighbors of different samples. **(D)** Novel pathways related to H1N1 VTP candidates. **(E)** Interaction network of relevant VTPs in the adherens junction pathway. **(F)** Distribution of novel, recovered, and missing labels. The classification is similar to that of VTPs. **(G)** Network topology and evolutionary conservation attributes of SVTPs and MVTPs. SVTP: single virus target protein, and MVTP: multiple virus target protein. **(H)** Top 10 biological processes enriched in MVTPs.

promise as potential antiviral targets. For instance, the non-VTP EMC8 (UniProt ID: O43402), predicted to interact with both DENV-2 and Zika virus (ZIKV), has been validated as an essential host factor for the replication of both viruses [50]. Another notable non-VTP, HSP90AB2P (UniProt ID: Q58FF8), was annotated as a target for six viruses by our model. Although we failed to validate its interactions with these viruses by literature mining, it has been identified as a therapeutic

target for SARS-CoV-2, supporting its potential as a target for different viruses [51]. The analyses for 11 virus families are presented in Fig Q in S3 File.

## Discussion

In this study, we developed a graph-based deep learning framework to identify VTPs using intrinsic information of host proteins. We first analyzed protein features and identified characteristics that differentiated VTPs from non-VTPs across multiple virus species/families. The analysis revealed that global network topological properties were the most important, followed by sequence conservation, and functional features could also provide valuable information. Based on this observation, we proposed MultiVTP, a multilabel algorithm that combined graph representation learning with multimodal data. Subsequently, ablation studies were conducted to validate the framework. At the feature level, global topological attributes were the primary contributors to performance, while multimodal features provided complementary information. At the module level, Graphormer was identified as the most critical component, while the PLE module also had a certain impact on performance. We further explored the interpretability of these two modules. The analysis showed that the graph transformer prioritized VTP-related proteins in the subgraph through higher attention scores, while the shared and specific experts in PLE module captured interaction patterns from complementary viewpoints. Extensive comparisons suggested that MultiVTP not only surpassed multiple baseline methods across both individual and overall performance metrics but also exhibited robustness under conditions of limited training data. In few-shot scenarios, MultiVTP demonstrated its potential for novel viruses that have few targets. More importantly, when applied to the human proteome, MultiVTP can predict novel VTP candidates for both individual and multiple viruses. This capability was further validated by our case studies on H1N1 and HIV-1, which demonstrated that VTP candidates could participate in various viral infection processes.

Despite the advantages of MultiVTP, two limitations were preliminarily identified in this study. One limitation is associated with the incompleteness of existing host-virus interaction databases. In this work, host proteins without documented interactions were treated as non-VTPs, which may lead to the introduction of false negatives. To address this issue, we integrated multiple databases and employed a more conservative strategy for non-VTP sample collection. The results further confirmed the robustness of our algorithm (Sections S1.12 and S1.13 in S1 File). The other limitation is that our algorithm is limited to assessing whether a host protein serves as a viral target. Given the availability of directional data on HIV-1 infection, we extended the model to predict the functional roles of host proteins during viral infection, such as identifying host factors essential for viral replication (HDFs) and those involved in antiviral responses (host restriction factors, HRFs). Our results implied that MultiVTP has the potential to predict these two classes of host factors (Section S1.14 in S1 File).

Additionally, several issues warrant further investigation. First, although the viral binding capability of VTPs depends on critical residues and functional domains, the current sequence embeddings generated by mean pooling lack positional information. Incorporating an attention mechanism to encode sequence information would not only optimize sequence representation but also enhance model interpretability [52]. Second, the discriminative capacity of multimodal features is limited in the current model, and simple concatenation of these features may fail to fully utilize the complementary information between different modalities. In the future, we could introduce an intra-modal contrastive loss to cluster VTPs from the same virus while separating non-VTPs, and adopt an attention-gated fusion module to adaptively fuse multimodal information to address these issues. Third, the model's performance on understudied viruses may be constrained by the highly imbalanced distribution of VTPs. This issue could be resolved by adopting a dynamically weighted loss function that assigns greater penalties to viruses with fewer VTPs [53]. Fourth, our method shows limitations in identifying VTPs targeted by a single virus, likely due to difficulties in learning virus-specific binding patterns. To address this, a two-stage learning strategy could be adopted, involving pre-training the model on the entire dataset followed by fine-tuning the specific expert of PLE module with the VTPs of individual viruses. Fifth, the current model does not incorporate protein

structural information. Since recent studies suggest that virus-targeted interfaces on host proteins often overlap with host PPI interfaces and exhibit higher evolutionary rates, integrating such structural insights could improve the performance of VTP identification [54]. Sixth, this study focuses exclusively on viral pathogens. In the future, the framework could be applied to other pathogens (e.g., bacteria and fungi) to predict pan-pathogen targeted host proteins. In summary, MultiVTP could be a useful tool for exploring VTPs across multiple viruses, thereby enhancing our understanding of host-virus interaction mechanisms.

## Conclusion

In this work, we presented MultiVTP, a multilabel framework for predicting VTPs by combining graph learning and multi-modal data. Specifically, we generated protein embeddings by jointly capturing their network topological properties and multimodal characteristics. A graph attention mechanism and a progressive hierarchical extraction module were then introduced to integrate and refine these feature descriptors, enabling the multilabel prediction of VTPs at both the virus species and family levels. Ablation experiments confirmed the critical role of graph structural features and the designed modules in improving performance, while other components further enhance prediction accuracy. MultiVTP consistently outperformed existing VTP and PPI prediction models across diverse experimental settings, with especially notable advantages when training data were scarce. Finally, we applied MultiVTP to the human proteome to systematically identify novel host proteins associated with single or multiple viruses.

## Materials and methods

### Datasets

In this study, we constructed two multilabel datasets, $D_{virus}$ and $D_{family}$, based on the virus species and virus families, respectively. To this end, we retrieved 20,422 manually annotated human proteins from the UniProt database [43]. By integrating human-virus PPIs from the HVIDB database, the above proteins were divided into 6,892 VTPs and 13,530 non-VTPs (proteins with no experimentally confirmed viral interaction) [55]. We adopted a two-step strategy to reduce sequence redundancy using CD-HIT, while preserving as many VTPs as possible [56]. In the first step, VTPs and non-VTPs were separately clustered using a 30% sequence identity threshold. Among each cluster, the longest sequence was retained as the representative, resulting in the non-redundant VTP set and non-VTP set. In the second step, these sets were combined and clustered using a 30% identity threshold. For clusters containing at least one VTP, the longest VTP was chosen as the representative; for clusters containing only non-VTPs, the longest sequence was reserved.
As a result, a total of 11,986 protein sequences were obtained for predictions. Each protein had a binary label vector $Y = \{y^1, \cdots, y^j, \cdots, y^n\}$, where $y^j = 1$ if there is a reported interaction between this protein and a virus (species or families) in the HVIDB database; otherwise, $y^j = 0$. To mitigate the imbalance of positive and negative samples, the virus having less than 200 VTPs was excluded from the training process of our main model (Section S1.15 in S1 File). Finally, the $D_{virus}$ dataset comprised 4,558 VTPs across 25 virus species and 6,856 non-VTPs, while the $D_{family}$ dataset consisted of 4,964 VTPs across 11 virus families and 6,856 non-VTPs. Both datasets were divided into training (80%) and testing (20%) sets.

### Subgraph sampling

PPIs play a fundamental role in biological processes, and the analysis of PPI networks is crucial for the understanding of infectious diseases. Here, we obtained experimentally validated human PPIs from the HIPPIE database [57]. After removing self-interactions and duplicates, we retained PPIs with confidence scores greater than 0 and constructed a human PPI network denoted as $\mathcal{G}(\mathcal{V}, \varepsilon)$, which consisted of 17,151 nodes and 703,919 edges. Since the neighborhood of a query protein may contain valuable information, we adopted the random walk (RW) algorithm to generate a

subgraph $g_u$ for each target node $u$. The RW started from the target node $u$ and proceeded for $l$ steps, which generated a subgraph of all nodes and edges visited in this walk. The transition probability between two consecutive nodes is defined as follows:

$$P_{u,v} = \frac{1}{d_u} \tag{1}$$

where $d_u$ represents the degree of node $u$ in the graph $\mathcal{G}$. To capture the different neighboring information, this procedure was repeated $c$ times, generating multi-view subgraphs $[g_u^1, g_u^2, \ldots, g_u^c]$. The parameters $l$ and $c$ were set to 20 and 3, respectively.

**Feature extraction**

**Network properties.** *Global topological properties.* The global topological role of each node (protein) was measured by applying the node2vec algorithm for embedding learning from the PPI network $\mathcal{G}$. Node2vec is a semi-supervised algorithm comprising two steps. First, for each node $u$ in $\mathcal{G}$, node2vec simulated biased random walks to generate node sequences $S_u$. The transition probability between nodes is defined as follows:

$$\alpha_{u,v} = \begin{cases} \frac{1}{p} & \text{if } d_{u,v} = 0 \\ 1 & \text{if } d_{u,v} = 1 \\ \frac{1}{q} & \text{if } d_{u,v} = 2 \end{cases} \tag{2}$$

where $p$ and $q$ are set to 4 and 1, respectively, and $d_{u,v}$ denotes the shortest path distance between nodes $u$ and $v$ (Section S1.16 in S1 File). Second, the generated node sequences were used to train a skip-gram model to optimize the $h^{global}$ through the following formulas, which maximized the log-probability of nodes in $S_u$ given the target node $u$ as follows:

$$\max \sum_{u \in \mathcal{V}} \sum_{v \in S_u} \log P(v|u) \tag{3}$$

$$P(v|u) = \frac{exp\left(h_u^{global} \cdot h_v^{global}\right)}{\sum_{v \in V} exp\left(h_u^{global} \cdot h_v^{global}\right)} \tag{4}$$

where $P(v|u)$ is the probability of node $v$ appearing in the node sequences, and $h_u^{global}$ and $h_v^{global}$ represent the feature representations of nodes $u$ and $v$, respectively. The algorithm was implemented using the Python libraries node2vec and gensim, producing a 256-dimensional feature vector [27]. For each subgraph $g_u^c$, its global topological properties were represented as $H_u^{(c)global} \in R^{l \times 256}$.

*Local topological properties.* For each sampled subgraph $g_u^c$, the shortest path distance between every pair of nodes $u$ and $v$ within this graph was computed using Dijkstra's algorithm. The distances were then encoded by a learnable function $b(u, v)$, resulting in a shortest path encoding matrix $SP_u^c \in R^{l \times l}$.

$$SP_{uv} = b(u, v) \tag{5}$$

**Multimodal features.** *Traditional features.* We quantitatively depicted each protein using different types of manually curated features. From the sequence perspective, we calculated the amino acid composition, which is the frequencies of 20 amino acids in a sequence. From the evolutionary perspective, three features were extracted, including the

homologous gene number, protein age and dN/dS ratio. Homologous gene number, representing the number of orthologous genes across species, was obtained from the HomoloGene database [58]. Protein age, reflecting the evolutionary history of a protein, was retrieved from the ProteinHistorian database [59]. dN/dS ratio was calculated from human-mouse orthologous genes using the Ensembl BioMart, indicating the selective pressure on protein-coding genes [60]. From the network perspective, we characterized each protein using four widely used graph features, including degree, closeness, betweenness, and clustering coefficient. From the structural perspective, the secondary structures (i.e., α-helix, β-strand, and coil) and solvent accessibility were predicted by the SPOT-1D-Single program [61]. Subsequently, we calculated four structural attributes for each protein, including the proportions of three secondary structure types and the average solvent accessibility.

Traditional features were concatenated into a vector $X^{TRA} \in R^{31}$. To refine the multifaceted information and prevent overfitting, an MLP was adopted to generate the traditional representation $x^1$.

$$x^1 = MLP\left(X^{TRA}\right) \tag{6}$$

*Sequence-based features.* Protein language models, trained on large-scale protein sequence data in an unsupervised approach, are capable of capturing rich biological information within sequences. We adopted the ESM2 model (esm2_t36_3B_UR50D) to encode protein sequences by extracting outputs from the last three hidden layers [40]. Each embedding has a size of $L \times 2560$, where $L$ is the sequence length.

The ESM2 embeddings were mean-pooled across residues to generate three 2560-dimensional vectors $X^{SEQ} \in R^{1 \times 2560}$. These vectors were then separately fed into three parallel MLPs. The outputs were concatenated and passed through a linear layer for dimensionality reduction, yielding the final sequence representation $x^2$.

$$x^2 = \left(MLP\left(X_1^{SEQ}\right) \parallel MLP\left(X_2^{SEQ}\right) \parallel MLP\left(X_3^{SEQ}\right)\right) W \tag{7}$$

where $\parallel$ denotes the concatenation operation, and $W$ is a learnable matrix.

*Functional features.* Proteins are typically associated with a group of functional terms [38]. In this study, we annotated each protein using the GO database and encoded its functional information with PubMedBERT, a language model pre-trained on large-scale biomedical literature [39]. Each GO term was processed by PubMedBERT, and the output of the pooling layer was retained. This generated a functional embedding matrix $X^{GO} \in R^{N \times 768}$, where $N$ denotes the number of GO terms.

Subsequently, we used the gosemsim program to compute similarities between the GO terms of each protein, thereby constructing a functional similarity graph [37]. We then adopted a GCN to aggregate information over the graph and employed mean pooling to derive the functional representation $x^3$.

$$x^3 = Pooling\left(GCN\left(X^{GO}\right)\right) \tag{8}$$

*Multimodal features.* For each protein in a subgraph $g_u^c$, three modality-specific representations $x^1$, $x^2$ and $x^3$ were concatenated to form a multimodal feature vector $x \in R^d$:

$$x = x^1 \parallel x^2 \parallel x^3 \tag{9}$$

where $d$ is the dimension of the feature vector. The multimodal feature matrix of a subgraph is denoted as $X_u^c \in R^{l \times d}$.

## Feature integration

Graphormer is built upon the Transformer architecture, which effectively integrates the self-attention mechanism with the topological information of networks [28]. In this study, we used Graphormer to integrate and upgrade the features of proteins within the multi-view subgraphs $[g_u^1, g_u^2, \ldots, g_u^c]$.

For each subgraph $g_u^c$, the multimodal feature matrix $X_u^c$ and global topological properties $H_u^{(c)global}$ were concatenated as the node features and fed into the attention module:

$$H_u^c = X_u^c \parallel H_u^{(c)global} \tag{10}$$

In the self-attention module, the matrix $H_u^c$, was projected using $W_Q$, $W_k$, and $W_v$, generating the corresponding representations Q, K, and V. Here, the attention scores reflected macro-level relationships among different proteins but failed to capture their relative positions in the network. Thus, local topological properties were incorporated as a bias to optimize the attention scores.

$$Q, \ K, V = LN\left(H_u^c\right) W_Q, \ LN\left(H_u^c\right) W_K, \ LN\left(H_u^c\right) W_V \tag{11}$$

$$H_u^{'(c)} = softmax\left(\frac{QK}{\sqrt{d_k}} + SP_u^c\right) V + H_u^c \tag{12}$$

where LN(·) is the layer normalization, $SP_u^c \in R^{l \times l}$ denotes the local topological properties of $g_u^c$, and $d_k$ is a scaling factor.

In addition to attention layers, Graphormer contains a fully connected layer after the self-attention module, which was applied to each protein separately and identically. This procedure consisted of two linear transformations and a ReLU activation.

$$H_u^{''(c)} = ReLU\left(LN\left(H_i^{'(c)}\right) W_1\right) W_2 + H_u^{'(c)} \tag{13}$$

where $H_u^{'(c)}$ is the output of self-attention module, and $W_1$ and $W_2$ are learnable matrices. The ReLU function was adopted to implement a non-linear transformation.

## Multilabel prediction

Modeling the binding patterns of VTPs for different viruses using a single model is challenging. Thus, we introduced a PLE module, which could separate shared and task-specific patterns explicitly and adopt a gradual routing mechanism to extract and separate more complicated information [29].

The updated representations of the query node $\left(\hat{h}_u^1, \hat{h}_u^2, \cdots, \hat{h}_u^c\right)$ were extracted from multi-view subgraphs $\left(g_u^1, g_u^2, \cdots, g_u^c\right)$, and were concatenated to construct the final embedding as follows:

$$\hat{h}_u = \hat{h}_u^1 \parallel \hat{h}_u^2 \parallel \cdots \parallel \hat{h}_u^c \tag{14}$$

The shared expert $e_{share}$ in PLE was responsible for capturing common patterns of VTPs across different viruses, whereas the specific expert $e_{specific}$ was designed to explore the specific characteristics of VTPs for each virus. In this study, 25 specific experts were constructed for the virus species dataset, while 11 specific experts were employed for the virus family dataset.

$$e_{share} = \hat{h}_u W_{share} \tag{15}$$

$$e_{specific} = \hat{h}_u W_{specific} \tag{16}$$

where $W_{share}$ and $W_{specific}$ denote the weight matrices of experts. The outputs of the shared expert and each specific expert were integrated through a gated mechanism for each prediction task. Notably, the parameters of a shared expert were influenced by all tasks, whereas those of specific experts were only affected by their corresponding task.

$$z_{specific} = softmax\left(\hat{h}_u W^g_{specific}\right)\left[\binom{e_{share}}{e_{specific}}\right] \tag{17}$$

where $W^g_{specific}$ denotes the gating matrix, and $\left[\binom{e_{share}}{e_{specific}}\right]$ represents the stacking of shared and specific experts. Finally, the integrated representation $z_{specific}$ was passed through an MLP to generate a prediction score for each virus.

$$\hat{y}_{specific} = Sigmoid\left(MLP\left(z_{specific}\right)\right) \tag{18}$$

where the sigmoid function converted the prediction values to a range of 0–1.

## Implementation and evaluation

We conducted 5-fold cross-validation on the training set to determine the optimal parameters of our algorithm and evaluated its generalization ability on the test set. The Adam optimizer was employed for parameter updates, with an initial learning rate of $5 \times 10^{-4}$. A learning rate warmup and decay strategy was adopted to improve training stability, where the learning rate linearly decreased from the initial value to zero. The binary cross-entropy was used as the loss function in the model optimization phase. All experiments were performed on an NVIDIA GeForce RTX 3080 Ti (12GB) GPU, with the training process of MultiVTP taking approximately 5 hours. We evaluated our model using several metrics, including recall, precision, F1-score, MCC, AUC, and AUPR. The performance of a model was evaluated using a weighted strategy, in which the weight of each label type was determined based on its frequency in the training set.

$$w^j = \frac{Num(j)}{\sum_{j=1}^{n} Num(j)} \tag{19}$$

$$I\left(\hat{y}_i^j \geq t^j\right) = \begin{cases} 1, & if\ \hat{y}_i^j \geq t^j \\ 0, & otherwise \end{cases} \tag{20}$$

$$Recall = \sum_{j=1}^{n} w^j \frac{\sum_{u=1}^{M} I\left(\hat{y}_u^j \geq t^j\right) * y_u^j}{\sum_{u=1}^{M} y_u^j} \tag{21}$$

$$Precision = \sum_{j=1}^{n} w^j \frac{\sum_{u=1}^{M} I\left(\hat{y}_u^j \geq t^j\right) * y_u^j}{\sum_{u=1}^{M} I\left(\hat{y}_u^j \geq t^j\right)} \tag{22}$$

$$F1 = \sum_{j=1}^{n} w^j \frac{2 * Recall^j * Precision^j}{Recall^j + Precision^j} \tag{23}$$

where $n$ is the total number of label types, $Num(j)$ is the number of target proteins of virus $j$ in the training set, and $M$ is the number of testing samples. $I(\cdot)$ is the standard indicator function, $t^j$ indicates the threshold for virus $j$, and $\hat{y}_u^j$ and $y_u^j$ are the predicted score and true label of protein $u$ for virus $j$, respectively.

## Supporting information

**S1 File. Supplementary experiments and results (Sections S1.1-S1.16).** This document contains the following sections: S1.1 Analysis of model specificity in predicting host-virus relationships. S1.2 Analysis of model specificity in predicting VTPs within shared pathways. S1.3 Impact of sample similarity on prediction performance. S1.4 Network dependency analysis of MultiVTP. S1.5 Effect of PPI data quality on model performance. S1.6 Evaluation of simplified MultiVTP architectures. S1.7 Interpretability analysis via linking attention weights. S1.8 Comparison of MultiVTP with virus-host PPI prediction methods. S1.9 Robustness assessment under extreme data scarcity and unseen viruses. S1.10 Generalization to novel viruses with limited samples. S1.11 Comparative analysis of HIV-1 VTP candidates and HDFs. S1.12 Model re-evaluation using an updated dataset. S1.13 Comparison of different negative sampling strategies. S1.14 MultiVTP extension for directional interaction prediction. S1.15 Sensitivity analysis of VTP count thresholds. S1.16 Reasons for various decisions in building our model.
(PDF)

**S2 File. Supplementary tables of this study (Tables A to G).** This document contains the following tables: Table A. Virus species and virus families in this work. Table B. Number of VTPs in the $D_{virus}$ dataset. Table C. Number of VTPs in the $D_{family}$ dataset. Table D. Number of VTPs in the few-shot learning dataset across 5-fold cross-validation. Table E. Performance of protein embeddings with different models. Table F. Novel VTP candidates in the main text and their supporting literature. Table G. Literature-supported novel pathways enriched in H1N1 VTP candidates.
(PDF)

**S3 File. Supplementary figures of this study (Figs A to Q).** This document contains the following figures: Fig A. Statistical analysis of traditional features of samples in the $D_{virus}$ dataset. Fig B. Statistical differences in traditional features between different sample groups in the $D_{virus}$ dataset. Fig C. Feature analysis and comparison on the $D_{family}$ dataset. Fig D. Statistical analysis of traditional features of samples in the $D_{family}$ dataset. Fig E. Statistical differences in traditional features between different sample groups in the $D_{family}$ dataset. Fig F. SHAP analysis of traditional features. Fig G. t-SNE visualization of global topological properties for non-VTPs (gray points) and VTPs from the $D_{virus}$ dataset. Fig H. Distribution of GO similarity among VTPs and between VTPs and non-VTPs for each virus. Fig I. t-SNE visualization of global topological properties for non-VTPs (gray points) and VTPs from the $D_{family}$ dataset. Fig J. Ablation experiments at the feature level. Fig K. Interpretability and ablation studies on the $D_{family}$ dataset. Fig L. Multilabel prediction strategies in machine learning. Figure M. Performance comparison of MultiVTP and baseline methods on the $D_{family}$ dataset. Fig N. Distribution of KEGG pathways enriched in predicted and known VTPs. Fig O. Analysis of HIV-1 VTP candidates. Fig P. Evaluation of our model using overlapping and virus-specific VTPs. Fig Q. Analysis of predicted and known VTPs in the human proteome across virus families.
(PDF)

## Author contributions

**Conceptualization:** Rong Liu.

**Data curation:** Kuang Ma, Kaiyu Liu.

**Formal analysis:** Kuang Ma, Kaiyu Liu, Yuhui Xin, Rong Liu.

**Investigation:** Kuang Ma, Kaiyu Liu, Yuhui Xin.

**Methodology:** Kuang Ma, Kaiyu Liu, Rong Liu.

**Resources:** Rong Liu.

**Software:** Kuang Ma.

**Supervision:** Rong Liu.

**Validation:** Kuang Ma, Kaiyu Liu, Yuhui Xin.

**Visualization:** Kuang Ma.

**Writing – original draft:** Kuang Ma, Kaiyu Liu.

**Writing – review & editing:** Rong Liu.

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
