## [Decision Letter · Decision Letter 0]

23 Feb 2026

PCOMPBIOL-D-25-02737

Multilabel Prediction of Virus Target Proteins via Multimodal Graph Representation Learning

PLOS Computational Biology

Dear Dr. Liu,

Thank you for submitting your manuscript to PLOS Computational Biology. After careful consideration, we feel that it has merit but does not fully meet PLOS Computational Biology's publication criteria as it currently stands. Therefore, we invite you to submit a revised version of the manuscript that addresses the points raised during the review process.

We look forward to receiving your revised manuscript.

Kind regards,

Nir Ben-Tal

Section Editor

PLOS Computational Biology

**Journal Requirements:**

At this stage, the following Authors/Authors require contributions: Kuang Ma, Kaiyu Liu, Yuhui Xin, and Rong Liu. Please ensure that the full contributions of each author are acknowledged in the "Add/Edit/Remove Authors" section of our submission form.

4) Please amend your detailed Financial Disclosure statement. This is published with the article. It must therefore be completed in full sentences and contain the exact wording you wish to be published.

State the initials, alongside each funding source, of each author to receive each grant. For example: "This work was supported by the National Institutes of Health (####### to AM; ###### to CJ) and the National Science Foundation (###### to AM).".

**Reviewers' comments:**

Reviewer's Responses to Questions

**Comments to the Authors:**

Reviewer #1: Comment 1: The focus of this research is on the interaction of viruses with human proteins, thereby infecting humans with certain diseases. There is no discussion about the factors that cause this interaction. For example, does the presence of identical motifs between the virus and the protein cause this interaction?

Comment 2: This research claims existing approaches remain time-consuming. It would be good to talk about the specific existing approach they talk about and show that the developed method outperforms those previously developed ones.

Comment 3: Briefly discuss what network topological properties you are talking about when claiming that your analysis revealed that specific network topological properties affect the interactions. Overall, you should be more specific in the whole discussions.

Comment 4: In the clustering of VTPs and non-VTPs you used a 30% sequence identity. Explain how the degree of sequence identity was measured. For example, did you use Needleman-Wunsch alignment method?

Comment 5: It is said that “we retained PPIs with confidence scores greater than 0”. In my opinion, keeping zero confidence score graph will increase the computation time of the whole method. Please provide a reason that you have to logically keep these graphs.

Comment 6: In general, the article does not explain the reasons for various decisions in many decision-making stages, for example, and are set to 4 and 1 respectively, using four structural attributes for each protein, adoption of ESM2 model, usage of GO database, and using the gosemsim program. It would be better if these reasons were clearly stated.

Comment 7: Although many papers are cited but some closely related papers such as “Alpha influenza virus infiltration prediction using virus-human protein–protein interaction network” are not recognized. It would be good to review such papers.

Comment 8: The implementation and evaluation section refers to many figures and diagrams that are included in the appendix of the article. However, it is not clear exactly what we have learned from this research. The article also lacks a conclusion section.

Comment 9: I had difficulty finding what star symbols refer to in Figure 2.

Reviewer #2: In this manuscript, the authors propose a multilabel deep learning framework named MultiVTP to identify virus target proteins (VTPs) using only intrinsic host protein information. The method integrates multimodal protein features with human protein–protein interaction networks and employs a graph transformer coupled with a progressive layered extraction (PLE) module to model both shared and virus-specific targeting patterns. In general, the work is solid, and the manuscript writing is fairly clear. However, some questions need to be addressed before it can be considered for publication. Below are specific comments and questions regarding the manuscript.

1. The necessity of VTP predictions. Although the authors explained the necessity of developing VTP predictions, this prediction task appears simpler than host-virus PPI predictions. In many cases, the interacting viral protein partners are even more important for deciphering the secret of host-virus relationships. The authors may wish to comment on this issue.

2. Another common prediction task in virus bioinformatics is to predict HDFs/HRFs (host dependent factors/host restrictive factors). What is the relationship between HDFs/HRFs and VTPs?

3. The non-VTPs are also a very important concept in this work. Therefore, the authors should provide a serious definition.

4. The manuscript defines host proteins without reported virus interactions in the HVIDB database as non-VTPs. However, this assumption does not adequately account for the severe incompleteness and reporting bias inherent in virus–host interaction databases. Many proteins labeled as non-VTPs may simply represent uncharacterized or insufficiently studied virus targets rather than true negatives. Moreover, HVIDB is not the only available resource for virus–host protein interactions. Other widely used databases, such as IntAct and BioGRID, also curate experimentally supported and predicted virus–host interactions. Restricting the definition of VTPs and non-VTPs to a single database may further amplify label noise and dataset bias.

5. In the section "Datasets", the authors mention that "To mitigate the imbalance of positive and negative samples, the virus having less than 200 VTPs was excluded from this study". However, later in the Results section, the authors claim that: "These results implied the potential of our model in few-shot learning tasks". Excluding viruses with fewer than 200 known targets introduces a strong selection bias toward well-studied viruses and directly contradicts the claim that the model is suitable for few-shot scenarios. Viruses with limited known targets are removed from the main dataset, yet the model is later presented as effective under limited-data conditions.

6. This inconsistency between dataset design and claimed application scope needs to be addressed. A clearer justification for the chosen threshold, or a sensitivity analysis demonstrating how model performance changes with different minimum VTP counts, would significantly strengthen the study.

7. The baseline comparisons in the manuscript are primarily limited to traditional machine learning models (e.g., XGBoost, SVM, Random Forest, and MLP) combined with standard multilabel learning strategies. While these comparisons are informative, they do not include recent state-of-the-art deep learning models for host–virus interaction or VTP prediction methods. Notably, several transformer- or GNN-based methods that leverage protein language models and network information are discussed in the Introduction, yet none of these approaches are included as baselines in the experimental evaluation.

8. The manuscript presents attention score distributions, t-SNE visualizations, and functional enrichment analyses to support model interpretability. However, it remains unclear how these analyses translate into concrete mechanistic insights. For example, while VTPs are shown to receive higher attention scores, it is not demonstrated whether these attention patterns correspond to specific biological functions, network motifs, or known host–virus interaction mechanisms. Additional analyses linking attention weights or learned representations to interpretable biological structures would strengthen the interpretability claims.

9. While the authors provide source code and model implementations, the manuscript lacks essential details regarding computational resources and training procedures. Information such as hardware configuration (e.g., GPU type and memory), training time, and scalability with respect to network size or number of viruses is not reported.

10. According to the current model architecture, can the proposed method be used for predicting VTPs for newly emerging viruses?

11. The analysis should follow the basic principle of biochemistry and human-virus biology. For example, the authors stated that “VTPs had more α-helices and fewer coils than non-VTPs, reflecting their potential structural specificity for interacting with the virus”. To prove this statement is reasonable, could the authors provide more evidence?

12. The prediction results showed that influenza A (H1N1) and HIV-1 had the highest number of candidates (2,555 and 1,681, respectively). Given the size of the human proteome, the predicted VTP number appears unexpectedly large (more than 10%). The authors may wish to comment on this issue.

13. Regarding the functional analysis of predicted VTPs, the authors may conduct a comparison of VTPs and HDFs (the HDFs for some viruses are known). Such a comparison may provide new hints to indirectly validate the prediction results.

Reviewer #3: 1. Scope and framing of the Introduction

The Introduction is generally well written but historically narrow and heavily method-focused. It would benefit from stronger grounding in field-defining experimental host–virus interactome studies.

Key examples that should be discussed include:

Jäger et al., (2011) — https://pubmed.ncbi.nlm.nih.gov/20708689/

Gordon et al., (2020) — https://pubmed.ncbi.nlm.nih.gov/32353859/

Stukalov et al., (2021) — https://pubmed.ncbi.nlm.nih.gov/33845483/

These foundational datasets underpin much of the training data used in computational host–virus PPI prediction and would better contextualise the importance of the problem.

2. Protein language models and foundation models

The manuscript does not adequately cover recent advances in protein language models (PLMs) and foundation models for interaction prediction.

For example:

Liu et al (2025) PLM-interact: Extending protein language models to predict protein–protein interactions - https://pubmed.ncbi.nlm.nih.gov/41145424/

This and related studies represent an important methodological shift and should be incorporated into the discussion of state-of-the-art approaches.

3. Directionality of virus–host interactions

The manuscript treats virus target proteins largely as a single functional class. However, host–virus interactions are biologically heterogeneous and include:

Pro-viral (host dependency factors)

Anti-viral (restriction factors)

The distinction is important both mechanistically and therapeutically. The authors already cite Chai et al. (PMID: 35134057) but do not introduce the complexity of the virus target proteins.

The authors should also discuss whether their framework can distinguish — or could be extended to distinguish — interaction directionality.

4. Methodological considerations: negative sampling

A key issue in PPI prediction that is not discussed is the construction of negative datasets.

Common challenges include:

Lack of experimentally validated true negative PPIs

Reliance on random protein pairing

Bias introduced into performance metrics

A good starting reference would be:

Ben-Hur & Noble, Bioinformatics (2006) — On negative examples in PPI prediction

This limitation should be acknowledged in the Methods or Discussion.

5. Acronyms and terminology

Several acronyms are not defined at first use. For clarity:

GCN — Graph Convolutional Network

GAT — Graph Attention Network

Please define all abbreviations when first introduced.

6. Cross-virus interaction overlap

The manuscript states:

“This result indicated that MultiVTP could better capture cross-virus interaction patterns.”

However, the extent of shared host targeting between viruses is not quantified.

It would strengthen the claim to report, for example:

Number of shared host proteins between virus pairs

Overlap coefficients / Jaccard indices

7. Figures and visualisation clarity

Several figures would benefit from improved labelling and legend placement.

General comment:

In many cases, colour legends or numeric virus identifiers are used where direct axis labelling with virus names would be clearer.

Specific suggestions:

Figure S1 — Difficult to interpret without x-axis labels

Figure S2 — Please add virus names on both x- and y-axes

Figure 2 (A–D) — Improve axis labelling clarity

Figure S8 — Clear and easy to interpret (good example)

Figure S5 — Current format acceptable given long family names

Improved labelling would substantially enhance accessibility.

8. Discussion of limitations

The Discussion should be expanded to address key limitations, including:

Inability to distinguish pro-viral vs anti-viral targeting

Potential biases in negative interactions

Virus coverage and generalisability

Reviewer #4: 1. The central premise of this study is the reformulation of virus target prediction as a multilabel task relying exclusively on host protein information. While methodologically appealing, this assumption requires substantially stronger biological justification. In many viral systems, targeting specificity is primarily governed by viral proteins that determine binding interfaces and interaction selectivity. By excluding viral determinants entirely, the proposed task may be closer to estimating host susceptibility rather than predicting genuine host–virus interactions. The authors should more clearly define the biological question their model is capable of addressing; otherwise, a mismatch may exist between the computational objective and the underlying molecular mechanisms. It is particularly important to clarify under which categories of host–virus interactions this assumption is expected to hold, and where it is likely to fail, especially for interactions mediated by virus-specific structural domains or accessory proteins.

2. The model demonstrates weaker performance for single-virus target proteins (SVTPs). This limitation appears biologically inevitable rather than algorithmic, as SVTPs are often driven by virus-specific molecular determinants that cannot be inferred from shared host characteristics alone. This observation raises concerns about whether the framework can truly capture mechanistically specific interactions rather than broad host vulnerability patterns.

3. Different viruses frequently converge on the same signaling pathways while employing fundamentally different molecular strategies. For example, two viruses may both target the NF-kB pathway, yet one may directly bind NFKB1 whereas another modulates upstream adaptor proteins. Without incorporating viral protein information, it remains unclear whether the proposed model can differentiate these mechanistic scenarios or merely capture pathway-level correlations.

4. The construction of negative samples warrants careful reconsideration. Proteins lacking documented interactions are treated as non-VTPs; however, host–virus interaction databases remain highly incomplete. As a result, the negative set likely contains undiscovered true targets, introducing label noise that may cause the model to learn patterns reflecting research intensity rather than biological reality. The authors should discuss this limitation more explicitly and consider alternative strategies such as positive–unlabeled learning or more conservative negative sampling.

5. Ablation experiments indicate that network topology contributes most strongly to predictive performance. This raises a critical concern that the model may primarily function as a detector of high-centrality proteins rather than identifying mechanistically relevant targets. Hub proteins in PPI networks are well known to be overrepresented in interaction datasets due to observational bias. If predictions are largely driven by centrality signals, the model may be capturing dataset artifacts rather than biological principles. Re-evaluating performance under degree-controlled settings would help clarify this issue.

6. Although the authors include few-shot experiments, viruses with extremely limited annotations (e.g., fewer than 20 validated targets) are not evaluated. This omission is non-trivial, as real-world deployment of such models is most critical precisely in scenarios involving emerging or poorly characterized viruses, where labeled data are inherently scarce. Consequently, the current experimental design may overestimate the practical utility of the framework. Moreover, the few-shot setting considered in this study still assumes the availability of a relatively informative training signal. It remains unclear whether the model can maintain stability when supervision becomes extremely sparse or highly imbalanced. Without testing under more stringent low-data conditions, the claim of robustness appears insufficiently supported. More challenging evaluation strategies, such as leave-one-virus-out validation or ultra-low-shot settings, would better approximate realistic application scenarios and demonstrate true generalizability.

7. Potential data leakage represents another important concern. Proteins targeted by multiple viruses frequently share sequence similarity, evolutionary origin, or network proximity. If homologous or topologically related proteins appear across training and testing splits with correlated label vectors, the model may benefit from implicit information transfer, thereby inflating predictive performance. While the manuscript mentions sequence redundancy reduction, it remains unclear whether redundancy was sufficiently controlled at the label level across folds. Standard clustering thresholds may not fully eliminate functional similarity, particularly within large protein families. Consequently, the reported performance may partially reflect memorization rather than genuine predictive capability. Providing a quantitative assessment of cross-fold homology, functional similarity, or neighborhood overlap would substantially strengthen confidence in the evaluation.

8. The exclusion of viruses with fewer than 200 annotated targets raises concerns about potential systematic bias. Well-studied viruses typically possess richer interaction data, more complete annotations, and stronger network signals, all of which may inadvertently simplify the prediction task. As a result, the model may become optimized for data-rich pathogens while exhibiting reduced reliability for understudied or newly emerging viruses. This filtering strategy may distort the empirical distribution of host–virus interactions and limit the ecological validity of the framework. In practice, computational tools are often most valuable precisely where experimental knowledge is limited. A model that performs well only on heavily characterized viruses may therefore have restricted translational relevance.

9. Regarding future work, please discuss the possibility of applying advanced learning techniques (10.1109/TCBBIO.2025.3610881 and 10.1109/JBHI.2025.3600045) for improved performance of the proposed model.

Reviewer #5: This manuscript presents MultiVTP, a multimodal graph-based deep learning framework for predicting virus target proteins (VTPs) using intrinsic host protein information. The problem addressed is important, and the multilabel formulation combined with multimodal graph learning is technically interesting. The manuscript is generally well organized and contains extensive computational analyses. However, several issues remain.

Major:

1. Dataset construction and potential information leakage

Because features include network topology, GO similarity graphs, and pretrained embeddings, proteins closely related to training examples may remain strongly coupled to them through the PPI network or functional similarity structure.

The authors should clarify whether homologous proteins or highly connected network neighbors are separated across splits, and whether alternative splitting strategies were explored. Without this clarification, performance estimates may be optimistic.

2. Benchmarking against recent methods

Comparisons focus largely on classical machine learning models and HIVPRE. Given recent advances in protein language models and deep-learning approaches for host–virus interaction prediction, additional contextualization would strengthen the manuscript. Either stronger comparisons or discussion of limitations in performing such comparisons would be helpful.

3. Model over(?)-complexity

The model combines:

• Traditional handcrafted biological features

• ESM2 sequence embeddings

• node2vec global topology embeddings

• Shortest-path distance encodings

• Graphormer-based attention

• A Progressive Layered Extraction (PLE) multilabel module

Individually, each component is reasonable. However, taken together, the architecture becomes quite layered and potentially redundant. Although the ablation studies show that removing certain components degrades performance, the reported improvements over strong baselines are moderate. This raises the question of whether a simpler architecture (e.g., ESM2 + node2vec + shallow GCN or shared multilabel MLP head) might achieve comparable performance with reduced complexity. Such a comparison would help clarify whether the proposed design is necessary.

Minor Comments

- The term “virus target protein (VTP)” is introduced without clear explanation in the abstract. A brief definition would improve accessibility for readers unfamiliar with this terminology.

- Some sections of the introduction could be shortened to improve focus.

**Have the authors made all data and (if applicable) computational code underlying the findings in their manuscript fully available?**

Reviewer #1: Yes

Reviewer #2: None

Reviewer #3: Yes

Reviewer #4: None

Reviewer #5: **No:** did not check

PLOS authors have the option to publish the peer review history of their article (what does this mean?). If published, this will include your full peer review and any attached files.

Reviewer #1: No

Reviewer #2: No

Reviewer #3: No

Reviewer #4: No

Reviewer #5: No

**Figure resubmission:**
---

## [Decision Letter · Decision Letter 1]

11 May 2026

Dear Dr. Liu,

We are pleased to inform you that your manuscript 'Multilabel Prediction of Virus Target Proteins via Multimodal Graph Representation Learning' has been provisionally accepted for publication in PLOS Computational Biology.

Please notice the very minor comment of Reviewer 3. You may revise the figure and submit the final draft.

Best regards,

Nir Ben-Tal

Section Editor

PLOS Computational Biology

Reviewer's Responses to Questions

**Comments to the Authors:**

Reviewer #2: Thanks for the efforts to address my previous comments.

Reviewer #3: Following the revisions, I appreciate the substantial effort the authors have invested in improving the manuscript. However, I still find several figures difficult to interpret. For example, in Figure A, it is challenging to reliably associate the 26 colours used in the legend with the corresponding boxplots. Additionally, the use of stacked bar plots to present AUPR values across different models makes comparative interpretation difficult. Overall, the manuscript would benefit from improvements to visualisations to more clearly facilitate readability and accessibility.

Reviewer #4: All of my concerns have been addressed in this revised version of the manuscript.

Reviewer #5: The authors have done a great job addressing the multitude of comments from the reviewers.

**Have the authors made all data and (if applicable) computational code underlying the findings in their manuscript fully available?**

Reviewer #2: Yes

Reviewer #3: Yes

Reviewer #4: None

Reviewer #5: Yes

PLOS authors have the option to publish the peer review history of their article (what does this mean?). If published, this will include your full peer review and any attached files.

Reviewer #2: No

Reviewer #3: No

Reviewer #4: No

Reviewer #5: No

---

## [Editor Report · Acceptance letter]

PCOMPBIOL-D-25-02737R1

Multilabel Prediction of Virus Target Proteins via Multimodal Graph Representation Learning

Dear Dr Liu,

I am pleased to inform you that your manuscript has been formally accepted for publication in PLOS Computational Biology. Your manuscript is now with our production department and you will be notified of the publication date in due course.

With kind regards,

Anita Estes
